# Effects of Preparation Parameters on the Structural and Morphologic Properties of SnO₂-Sb Coatings

Zhen He ⬤, Chen Yu, Jiaming Liu, Zengcheng Miao and Yuxin Wang *⬤

School of Materials Science and Engineering, Jiangsu University of Science and Technology,
Zhenjiang 212003, China
* Correspondence: ywan943@163.com

**Abstract:** Sb-doped SnO₂ (SnO₂-Sb) coatings show superiority in degrading toxic and refractory organic pollutants. SnO₂-Sb coatings can be prepared by oxidizing electrodeposited Sn-Sb coatings through an annealing process. The properties and structure of SnO₂-Sb coatings can be tailored by adjusting the preparation parameters. This study examines the effects of crucial preparation parameters on the performance of SnO₂-Sb coatings, with the aim of enhancing their properties. Determining the coatings' cross-sectional and surface characteristics was accomplished using various characterization techniques. A thorough investigation of the prepared samples' phase and element components was also carried out. Based on the findings, the surface roughness of the prepared Sn-Sb precoating changed with increasing current density, yet the primary surface features of the SnO₂-Sb coatings were hardly altered by the annealing process. Without lowering the coating thickness, the appropriate current density of 30 mA/cm² produced a rough and active coating surface. Our study's proper annealing temperature of 600 °C transformed Sn-Sb precoating into SnO₂-Sb coating and achieved excellent coating quality. While changes in the Sb content affected the morphology of the prepared SnO₂-Sb coatings, the mixed oxide coatings' cassiterite SnO₂ phase was unaffected. These results may provide insights into the optimized use of SnO₂-Sb coatings in various applications.

**Keywords:** SnO₂-Sb; electrodeposition; structures

## 1. Introduction

Environmental pollution caused by industrial waste is regarded as a significant issue. The use of electrocatalytic oxidation for the treatment of refractory effluents has been extensively investigated along with rapid industrialization worldwide [1–3]. The anode material plays a vital role in electrocatalytic oxidation reactions, which determines the oxidation efficiency and final oxidation products [4,5]. Given its stable size, long life, high electrocatalytic performance, and low cost, the dimensional stable anode (DSA) electrode is a promising electrocatalytic anode material. Among the DSA electrodes, the titanium-based Sb-doped SnO₂ (Ti/SnO₂-Sb) electrode has shown superiority in degrading various organic pollutants, especially in treating toxic organic pollutants and some refractory organic pollutants [6–8].

So far, multiple preparation methods, including thermal decomposition, electrodeposition, and chemical vapor deposition, are available to obtain SnO₂-Sb coatings on a Ti substrate [9,10]. Although thermal decomposition has been used extensively to make SnO₂-Sb coatings, the complex procedures make preparation challenging, and its open structure results in an inferior service life. In contrast, the electrodeposition technique has recently been developed by researchers to create SnO₂-Sb coatings [11]. Electrodeposition demonstrates its high efficiency and compatibility in coating preparation. For instance, Ding and co-authors produced SnO₂-Sb coatings on a titanium substrate by cathodic electrodeposition and annealing, which revealed a cluster structure on the coatings' surface. Wu and colleagues created SnO₂-Sb coatings on TiO₂ nanotubes using pulse electrodeposition to achieve a solid electrochemical oxidation ability and high oxygen evolution potential.

$SnO_2$-Sb coatings' structure and properties could be carefully tailored by altering their preparation parameters [12–14]. Subtle variations in the electrodeposition process affect the resulting properties, and preparation parameters determine the structure and components of the coatings produced. When preparing $SnO_2$-Sb by the electrodeposition method, many crucial preparation parameters are present, including Sb concentration, current density, and annealing temperature [15–17]. For example, Qing Ni investigated the effects of Sb content on the degradation performance of $SnO_2$-Sb coatings prepared by the electrodeposition method [18]. Despite this, there are intricate correlations between coating properties and preparation conditions, as these parameters are not independent of each other [19,20]. Such correlations have been examined extensively, in which the outcomes from different studies may sometimes not be comparable due to the various preparation processes. The critical characteristics of prepared coatings, such as active areas and long-term stability, are directly determined by preparation parameters such as current density and annealing temperature. Therefore, in-depth research is still very much needed on the precise impact of these crucial preparation parameters on $SnO_2$-Sb coatings [21–24].

This work successfully prepared $SnO_2$-Sb coatings on Ti substrates by the electrodeposition method from a sulfate medium [25]. This study attempts to demonstrate a clear correlation between the essential parameters and characteristics of $SnO_2$-Sb coatings. The effects of Sb concentration, current density, and annealing temperature on the structure and constituents of the coatings produced were comprehensively investigated. This study's findings could advance the understanding of $SnO_2$-Sb electrodeposition preparation and pave the way for improving its electrochemical performance in the future.

## 2. Experiments and Materials

### 2.1. Pretreatments of Ti Substrates

Pure Ti sheets, which were 15 mm × 15 mm ×2 mm in size, were used in this study. The chemicals used were supplied by Aladdin Co. (Shanghai, China) as analytical reagents and used as received. Grounding and sandpaper polishing up to a 2000 mesh were used to finish the titanium sheet. The polished titanium sheet was placed in a 15 wt.% NaOH aqueous solution at 85 °C for 2 h at a constant temperature. The alkaline treated titanium sheet was immersed in a 12 wt.% oxalic acid solution at 85 °C for 3 h, followed by ultrasonic washing for 10 min.

### 2.2. Preparation of Sn-Sb Precoatings

By electrodeposition of the cathode current in a sulfate electroplating medium, $SnO_2$-Sb coatings were produced on pretreated Ti substrates. The sulfate electrolyte consisted of 65 mL/L $H_2SO_4$ (sulphuric acid), 10 g/L $SnSO_4$ (stannic sulfate), 0.25~0.75 g/L $SbCl_3$ (antimony chloride), 2 g/L $C_{12}H_{16}O_{11}$ (gelatin), and 1 g/L $C_6H_6O_2$ (p-dihydroxybenzene). The ratios of Sb/Sn in atomic percentages in the electrolytes were 2.5%, 5.0%, and 7.5% in the electrodeposition process. A pretreated titanium sheet served as the cathode and a tinplate (40 mm by 40 mm) served as the anode for the electrolytic cell. At a specific current density (10, 20, 30, 40, and 50 mA/cm$^2$), cathodic electrodeposition was performed at a stirring speed of 300 rpm. The electroplating time was 10 min and the temperature was 30 °C. After the cathodic electrodeposition, the Sn-Sb precoating was prepared on the Ti substrate.

### 2.3. Preparation of $SnO_2$-Sb Coatings

Further oxidation treatment was necessary to obtain the $SnO_2$-Sb coatings. The as-prepared Sn-Sb precoatings were transferred into oxides in a muffle furnace for annealing treatment at certain temperatures (500, 600, 700 °C) for 10 h, followed by air cooling to obtain the $SnO_2$-Sb coatings.

*2.4. Characterization Tests*

A scanning electron microscope (SEM, ProX, Phenom; Eindhoven, The Netherlands) was used to determine the coatings' surface and cross-sectional structures, and an embedded energy dispersive spectrometer (EDS) detector was used for elemental analysis. The phase components were identified using an X-ray diffraction (XRD-6000X; Shimadzu, Kyoto, Japan) machine with Cu Ka radiation (Rouse = 1.54056 Ü).X-ray photoelectron spectroscopy (XPS, ESCALAB 250Xi; Thermo Fisher Scientific, Waltham, MA, USA) was utilized to analyze elemental details.

## 3. Results and Discussion

*3.1. Effects of Current Density*

Figure 1 shows the surface morphologies and elemental compositions of the electrodeposited Sn-Sb precoatings and annealed $SnO_2$-Sb coatings. The prepared Sn-Sb precoatings' surface features were smooth and dense at low current densities (10 and 20 mA/cm$^2$), as depicted in Figure 1a,c. As the current density increased to 30 mA/cm$^2$, tiny pores began to appear on the precoating surface, resulting in a significant increase in surface roughness (Figure 1e). However, when the current density increased to 50 mA/cm$^2$, the Sn-Sb precoatings revealed a non-uniform surface morphology, making the coating quality inferior as shown in Figure 1i.

The surface characteristics of the $SnO_2$-Sb coatingswere hardly altered compared to the Sn-Sb precoatings, even though the precoatings were oxidized; their surface morphology was largely preserved. It can be seen that the $SnO_2$-Sb coatings had relatively flat surface features when prepared at 10 and 20 mA/cm$^2$, and their coating surface was dense and had few pores as indicated in Figure 1b,d. When the current density reached 30 mA/cm$^2$ as shown in Figure 1f, the $SnO_2$-Sb coatings exhibited a cluster-like surface structure, where the small crystallites coalesced into numerous surface clusters. However, by increasing the current density to 40 and 50 mA/cm$^2$, as shown in Figure 1h,j, the agglomeration process became more evident and prominent pores appeared. In the meantime, the surface clusters became smaller and their number increased.

Polarization behavior could explain the variations in surface morphology at the increasing current densities during the electrodeposition process [26]. Precisely, the polarization effect was enhanced due to the high overpotential caused by the high current density, during which the side reaction of hydrogen evolution was intensified. The release of hydrogen gas created surface pores and protrusions, significantly altering the surface morphology of the precoatings.

The attached elemental data support the preparation of the precoatings and $SnO_2$-Sb coatings [27]. Sn dominated the prepared precoatings, but it is noteworthy that an increasing current density resulted in higher Sb content in the Sn-Sb precoatings. This variation in elemental composition occurred due to the different reduction potentials of Sn and Sb. The increased Sb content during precoating at higher current densities could be attributed to a greater reduction overpotential of the Sb electrodeposition reactions [28].

Figure 2 shows the XRD diffractograms of the Sb-Sn precoatingsand $SnO_2$-Sb coatings prepared at different current densities. The positions of the diffraction peaks are consistent with the standard Sn (JCPDS:04-0673) and Sb (JCPDS:33-0118) diffractograms, which proves the successful electrodeposition of Sn and Sb on the Ti substrates. In all samples, the dominant peaks originated from Sn. It should be noted that the increased current density reduced the intensity of the most significant Sn diffraction peaks, which indicates the refined grain size under the high current density. It has extensively been reported that small grain sizes are correlated with the high overpotential at a high applied current density [29,30]. Inaddition, a higher current density above 30 mA/cm$^2$ led to a higher content of Sb in the precoatings, as indicated by more Sb peaks recorded in the profiles, which agrees with previous EDS data.

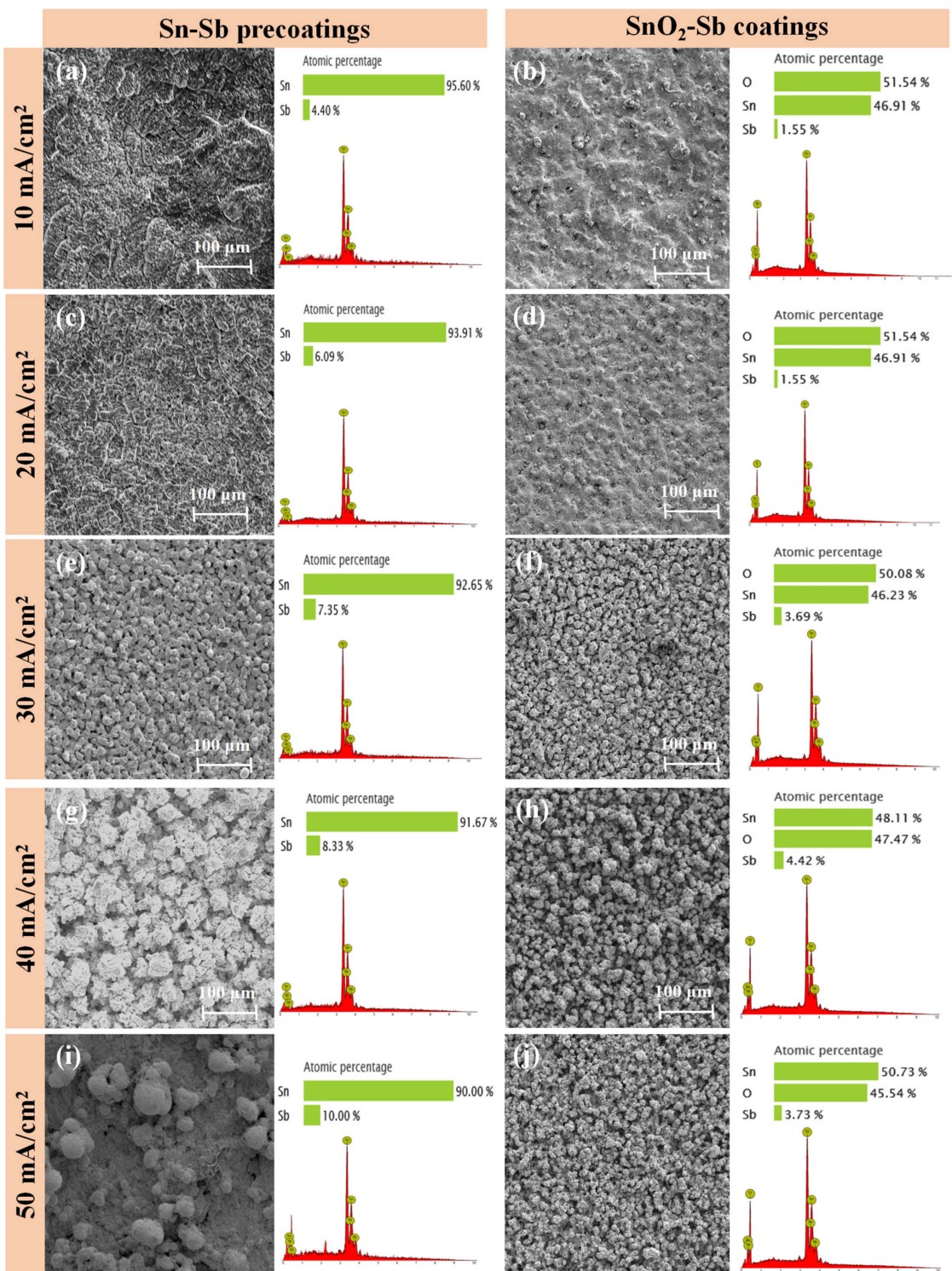

**Figure 1.** Surface morphologies and elemental compositions of Sn-Sb precoatings prepared at different current densities: (**a**) 10 mA/cm$^2$, (**c**) 20 mA/cm$^2$, (**e**) 30 mA/cm$^2$, (**g**) 40 mA/cm$^2$, (**i**) 50 mA/cm$^2$. Morphologies and elemental compositions ofSnO$_2$-Sb coatings prepared at different current densities: (**b**) 10 mA/cm$^2$, (**d**) 20 mA/cm$^2$, (**f**) 30 mA/cm$^2$, (**h**) 40 mA/cm$^2$, (**j**) 50 mA/cm$^2$.

Despitethe XRD pattern differences for the precoatings under different current densities, the annealed SnO$_2$-Sb coatings show similar phase constituents for all samples (Figure 2b). Most diffraction peaks belonged to SnO$_2$, while no peaks from the antimony oxide were detected in the coatings.

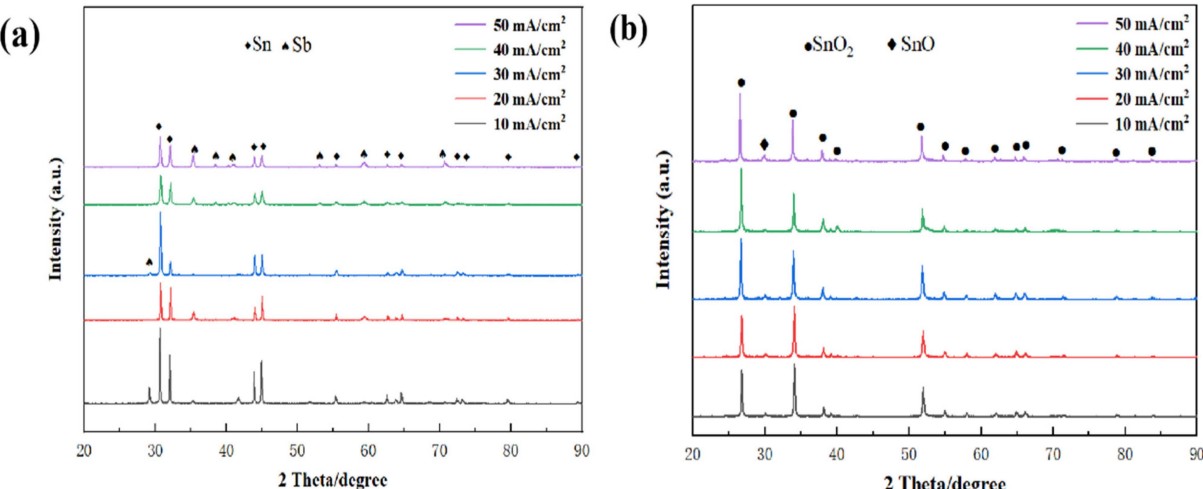

**Figure 2.** XRD diffractograms recorded of (**a**) Sn-Sb precoatings and (**b**) $SnO_2$-Sb coatings prepared at different current densities.

The cross-sectional morphologies of the samples prepared by different current densities were also determined, as depicted in Figure 3. The $SnO_2$-Sb layer was electrodeposited on the Ti substrates for all samples. When the current density was below 30 mA/cm$^2$, the prepared $SnO_2$-Sb coating was well-coated, fully covering the substrate with a good quality. The estimated thickness of the $SnO_2$-Sb coating increased from ~14 um at 10 mA/cm$^2$ to ~23 um at 30 mA/cm$^2$.

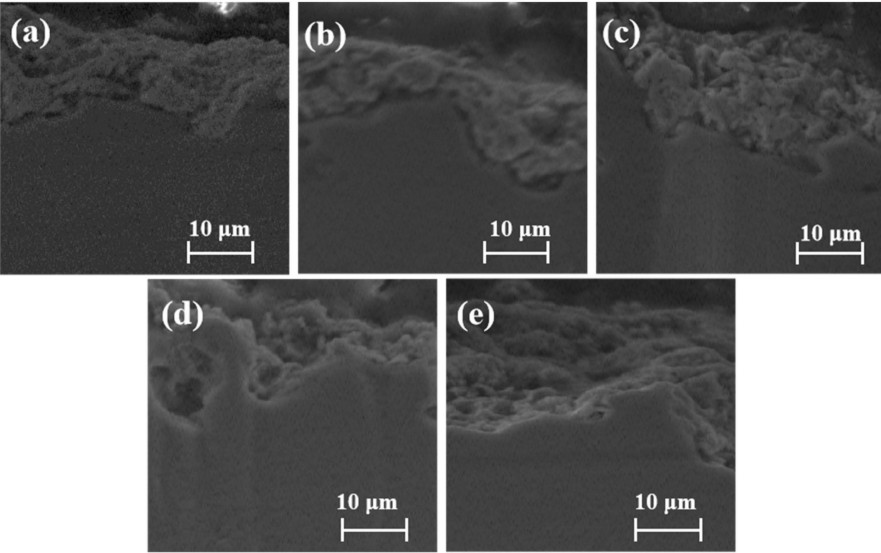

**Figure 3.** Cross-sectional morphologies of $SnO_2$-Sb coatings prepared at different current densities: (**a**) 10 mA/cm$^2$, (**b**) 20 mA/cm$^2$, (**c**) 30 mA/cm$^2$, (**d**) 40 mA/cm$^2$, (**e**) 50mA/cm$^2$.

When the current density was higher than 30 mA/cm$^2$, a notable morphologic change trend could be noted: the higher the applied current density, the thinner the coated $SnO_2$-Sb coating (i.e., 5~10 um). The observed phenomena match the surface observations of the samples, which suggests that an open coating surface correlates with a small coating thickness. To further verify such phenomena, we carried out the cross-sectional images for typical samples using a different preparation method. Specifically, we cut the plate sample directly and performed SEM observation without polishing or grounding, which helped to avoid the possible detachment of a porous coating. We found a consistent trend in that an excessively high current density resulted in a thinner coating and a relatively porous surface structure, as shown in Figure S1 in the Supplementary Materials.

Under a high current density, the intense hydrogen evolution consumedmuch of the charge in the electrodeposition process. The release of hydrogen gas gives an open surface and leads to a thinner coating. The subsequent annealing treatment hardly changed such variation trends.

XPS measurements were carried out to analyze the chemical state of the surface elements for the prepared $SnO_2$-Sb coatings, as shown in Figure 4. Figure 4a shows the overall spectrum. The annealing treatment oxidized the precoating into its oxide form. Figure 4b–d revealed that $Sb^{3+}$ and $Sb^{5+}$ were present in the $SnO_2$-Sb coatings, corresponding to the two oxidation states for Sb that existed in the $SnO_2$ lattice. Moreover, $O_{lat}$ (i.e., lattice oxygen species) and $O_{ads}$ (hydroxyl oxygen species) are the two oxygen species that were identified, consistent with previous research [31].

On the basis of the XPS analysis, Table 1 illustrates the calculated ratios of $Sb^{5+}/Sb^{3+}$ and $O_{ads}/O_{lat}$. The higher proportion of $Sb^{5+}/Sb^{3+}$ indicates that there was an adequate oxidation reaction during annealing. Generally speaking, $Sb^{5+}$ is the prominent electron donor for $SnO_2$-Sb coatings. The current density of $30\ mA/cm^2$ generated a more conductive coating with a higher proportion of $Sb^{5+}/Sb^{3+}$. Increased $Sb^{5+}$ content gave rise to improved conductivity and an active surface. The lattice oxygen was reduced at the same time as the natural suppliers of oxygen vacancies were suppressed.

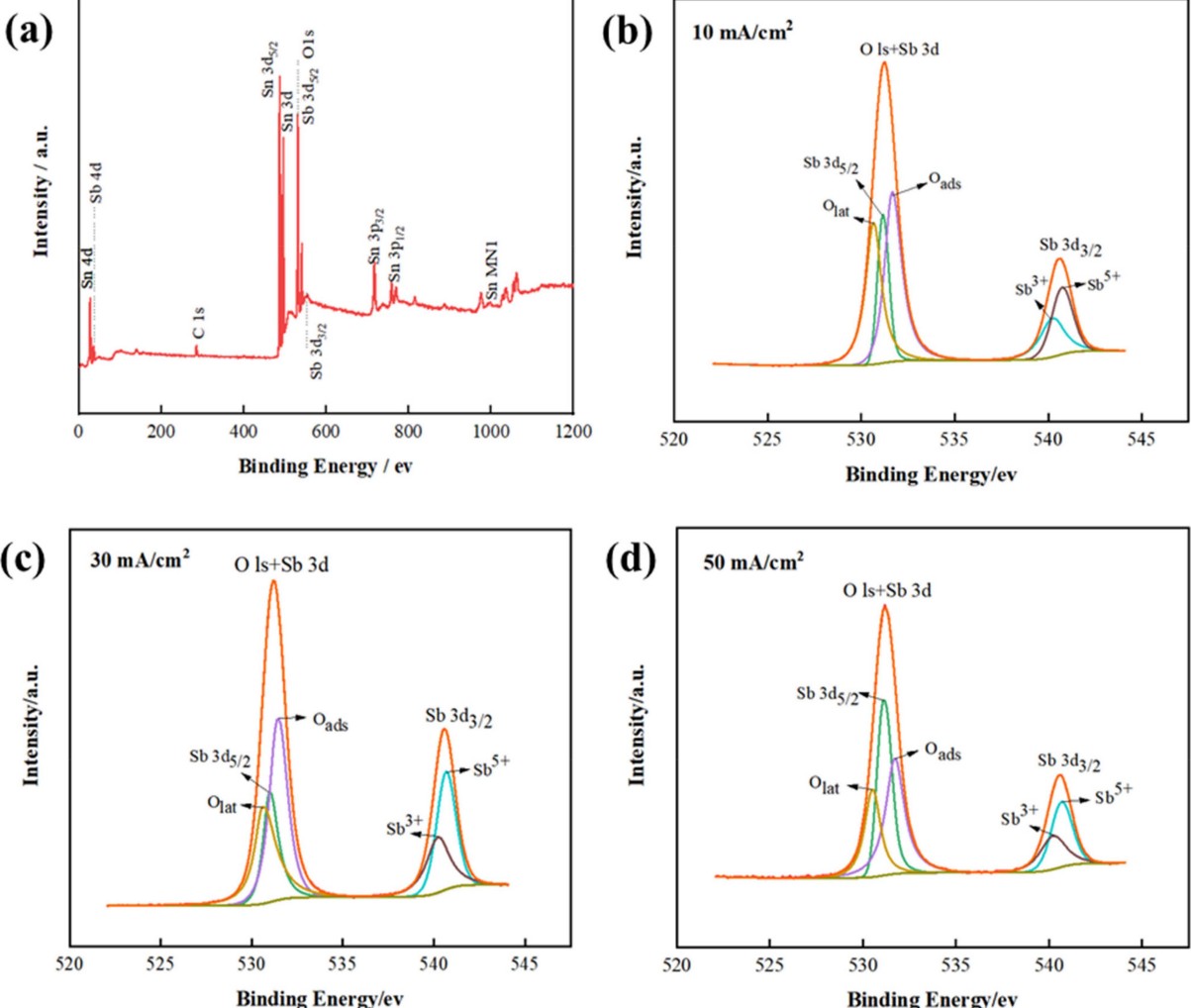

**Figure 4.** XPS spectra recorded of samples prepared at different current densities: (**a**) full spectrum; (**b**) O and Sbspectra detected of the samples prepared with the current density of $10\ mA/cm^2$, (**c**) $30\ mA/cm^2$, (**d**) $50\ mA/cm^2$.

**Table 1.** Calculated atomic ratios for the samples prepared at different current densities.

| Applied Current Density | AtomicRatio | |
|---|---|---|
| | $Sb^{5+}/Sb^{3+}$ | $O_{ads}/O_{lat}$ |
| 10 mA/cm$^2$ | 1.24 | 0.53 |
| 30 mA/cm$^2$ | 1.52 | 0.72 |
| 50 mA/cm$^2$ | 1.36 | 0.61 |

The surface characteristics of SnO$_2$-Sb coatings, such as roughness, conductivity, and active areas, are essential to determine for their potential applications, such as electrochemical catalysis and energy storage. Furthermore, the coatings' complete coverage of the substrate used is a premise in its applications. In this regard, it is possible to subtly customize SnO$_2$-Sb coatings' surface morphology for various usages by varying their electrodeposition parameters. In this regard, our study could be a reference for related research.

### 3.2. Effects of Annealing Temperature

Theeffects of annealing temperature on the oxide layers were also comprehensively investigated. The experiments used a current density of 30 mA/cm$^2$, considering their rough surface and good coating quality. Heat treatment temperatures of 500, 600, and 700 °C were applied to prepare the SnO$_2$-Sb$_2$O$_x$ coatings.

Figure 5 presents the surface information of the oxide coatings. It should be noted that the annealing temperature of 500 °C could not fabricate a rough surface morphology, as indicated in Figure 5a,b, where a relatively flat surface consisting of rectangular crystals was observed for the SnO$_2$-Sb coatings. In contrast, a higher heating temperature of 600 °C gave a distinct surface morphology showing numerous cauliflower-shaped crystals. Further increasing the temperature to 700 °C turned the coating into a relatively non-uniform surface made up of separate spheres.

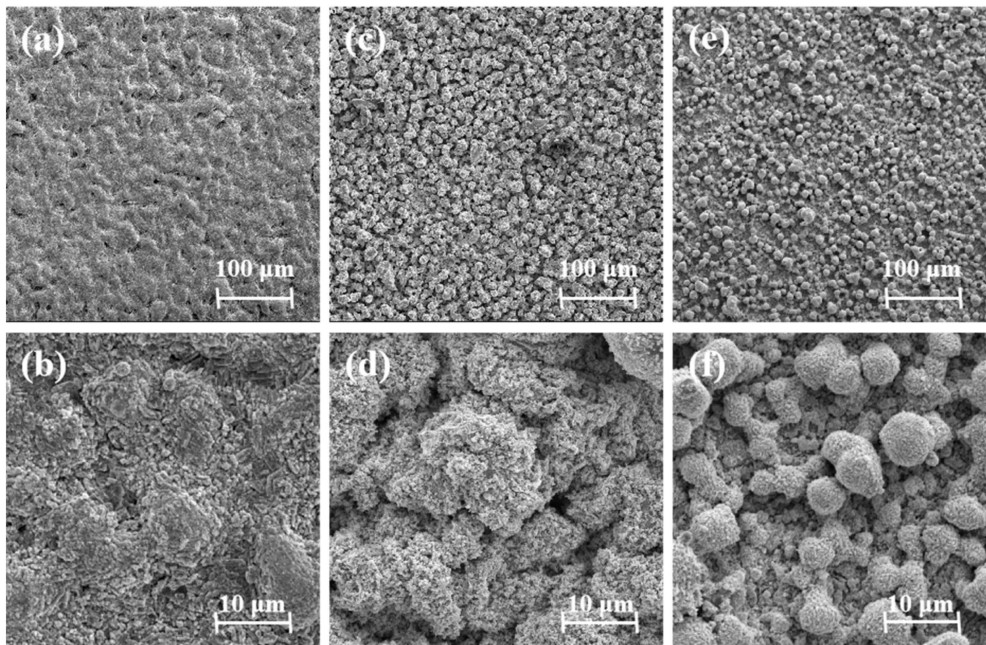

**Figure 5.** Surface morphologies and magnification plots of SnO$_2$-Sb coatings prepared at different annealing temperatures: (**a**) 500 °C, (**b**) 500 °C magnification, (**c**) 600 °C, (**d**) 600 °C magnification, (**e**) 700 °C, (**f**) 700 °C magnification.

The phase constituents for the samples prepared at different heating temperatures are presented in Figure 6. At the temperature of 500 °C, sharp $SnO_2$ peaks were detected, yet a minor Sn peak still existed in the coatings. This indicates an inadequate oxidation process at a low temperature, which harmed the coating performance. The annealing treatment under an increased temperature entirely changed the coatings into their oxide form. It sharpened the $SnO_2$ diffraction peaks due to the promoted grain growth process at a higher oxidation temperature.

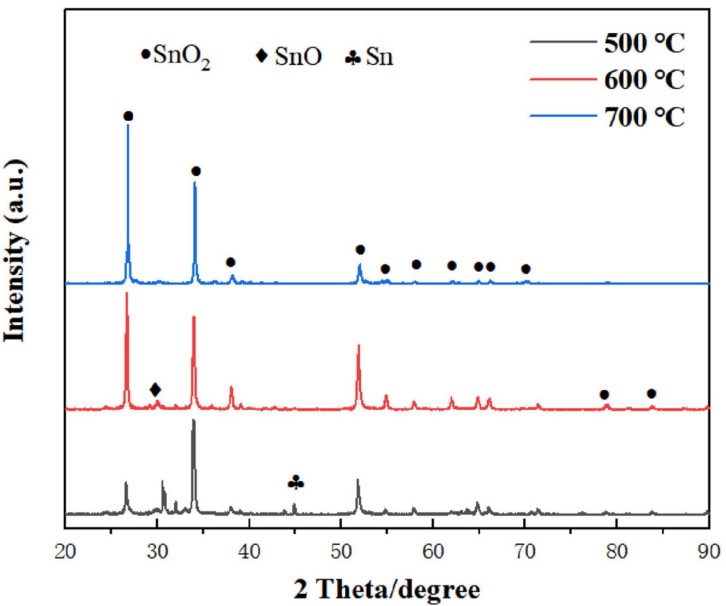

**Figure 6.** XRD patterns recorded of the $SnO_2$-Sb coatings prepared at different annealing temperatures.

Cross-sectional information was recorded and is displayed in Figure 7. The annealing treatment at a low temperature of 500 °C could not transform the precoatings into their oxide form, and a limited coating thickness of ~13 um was observed, as shown in Figure 7a. The annealing treatment at 600 °C resulted in a much thicker and uniform coating with a thickness of ~23 um, as depicted in Figure 7b. However, under an excessively high temperature at 700 °C, the thickness distribution of the oxide coating became much less uniform, as shown in Figure 7c, which agrees with the surface morphologic observations above.

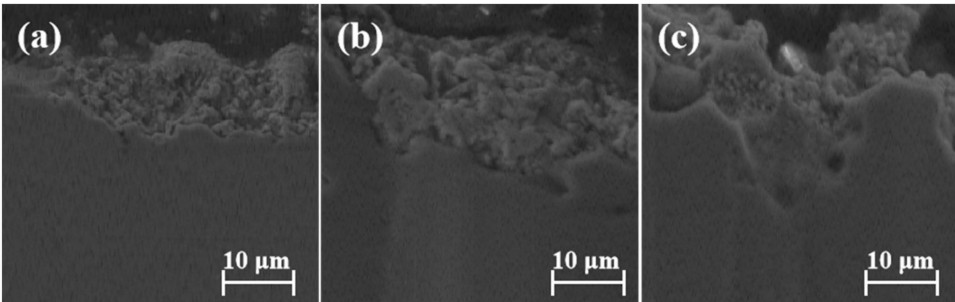

**Figure 7.** Cross-sectional morphologies of $SnO_2$-Sb coatings prepared at different annealing temperatures: (**a**) 500 °C, (**b**) 600 °C, (**c**) 700 °C.

The annealing temperature alters the oxidation process significantly [32–34], during which Sn-Sb precoatingsare transformed into $SnO_2$-Sb coatings. As proposed by our study, selecting a proper annealing temperature is of great importance: a low temperature cannot fully oxidize the precoatings and a high temperature causes severe non-uniformity. In contrast, our study's proper annealing temperature of 600 °C preparedSnO_2-Sb coatingsof an excellent quality.

### 3.3. Effectsof Sb Concentration on Electrolytes

This research also investigated the effects of Sb concentration on the electrolytes within the prepared coatings. Three different concentrations of 2.5%, 5.0%, and 7.5% (Sb: Sn in atomic percentagesof electrolytes) were selected for the experiments. Figure 8 presents the surface morphologies and elemental results recorded on the Sn-Sb precoatings and $SnO_2$-Sb coatings. A small concentration at 2.5% caused a relatively flat surface morphology with some spherical crystals. When a 5.0% Sb concentration was applied, more rough and open surface features were attained. However, upon further increasing the Sb concentration to 7.5%, a flatter surface morphology was obtained for the precoating, causing a non-uniform surface morphology for the oxide coating after the annealing treatment, as shown in Figure 8f.

In addition, the EDS results presented in Figure 8 indicate that the increasing Sb concentration in the electrodeposition directly resulted in higher Sb content in the prepared coatings. It should be noted that the Sb content in the Sn-Sb coatingswas higher than in the electrolytes, resulting from the higher overpotential of Sn reduction reactions [35].

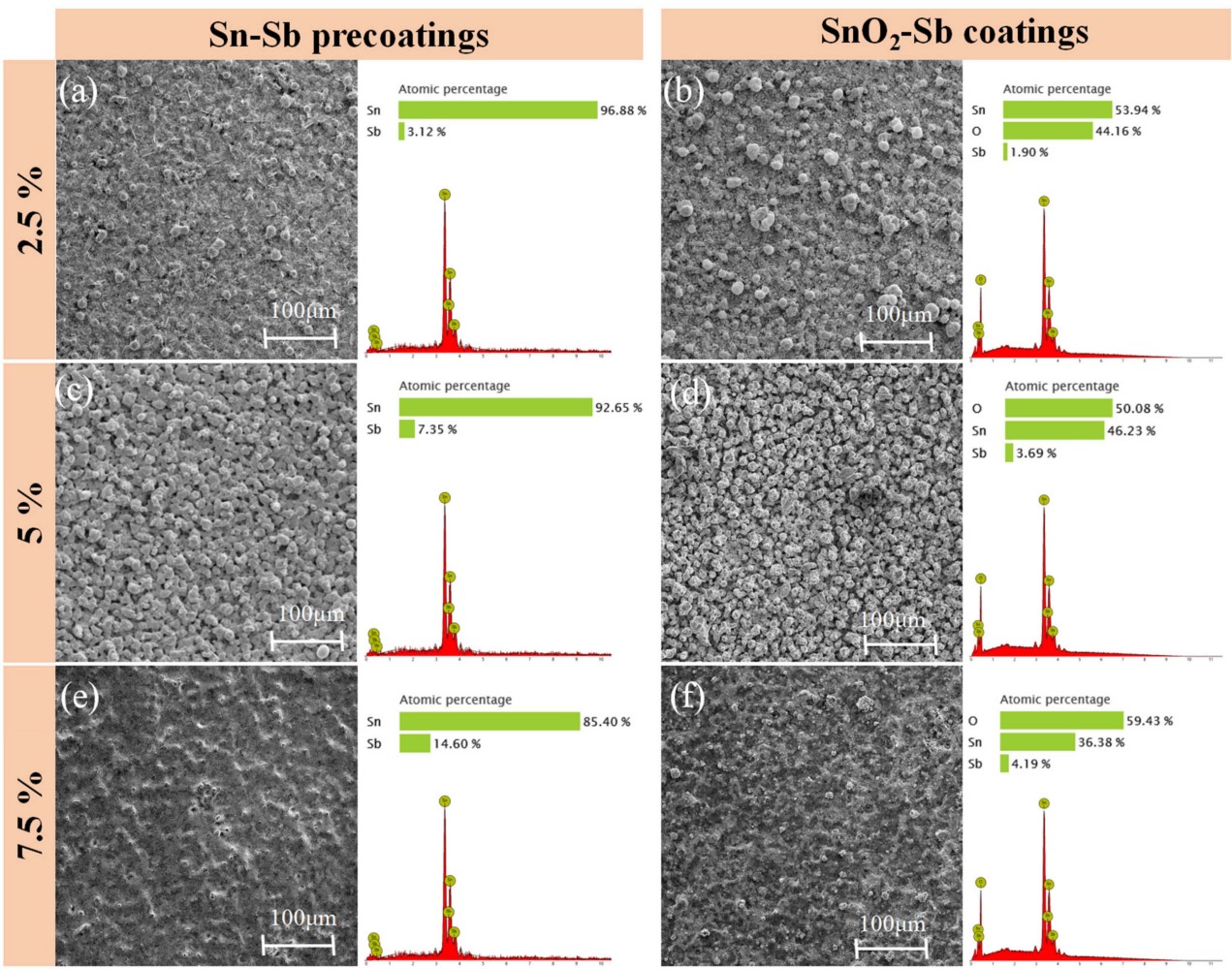

**Figure 8.** Surface morphologies and elemental compositions of Sn-Sb precoatingsandSnO₂-Sb coatings prepared at different Sb concentrations: (**a**) 2.5%, (**c**) 5.0%, (**e**) 7.5%.Morphologies and elemental compositions ofSnO₂-Sb coatings prepared at different Sb concentrations: (**b**) 2.5%, (**d**) 5.0%, (**f**) 7.5%.

The phase constituents of the prepared samples are presented in Figure 9. The Sn-Sb precoatings were electrodeposited on the Ti substrates for all samples, as shown in Figure 9a. The high Sb content in the electrolytes gave rise to several Sb peaks appearing for the Sn-Sb precoatings. Interestingly, the increased Sb concentration gave no $Sb_2O_x$ diffraction peak

in the patterns, indicating that the Sb was doped by substituting the Sn in the lattice. The activity of mixed oxides was significantly affected by Sb doping. In the case of pure $SnO_2$, the non-conductivity made them unsuitable anode materials. The phase composition of the $SnO_2$-Sb coatings may be influenced by the doping level, which in turn, impacts electrical conductivity. The cassiterite $SnO_2$ was present in the mixed oxide. By substituting $Sn^{4+}$ ions with doped Sb at their locations, the extra valence electrons in Sb acted as donors and $SnO_2$ was more conductive [36].

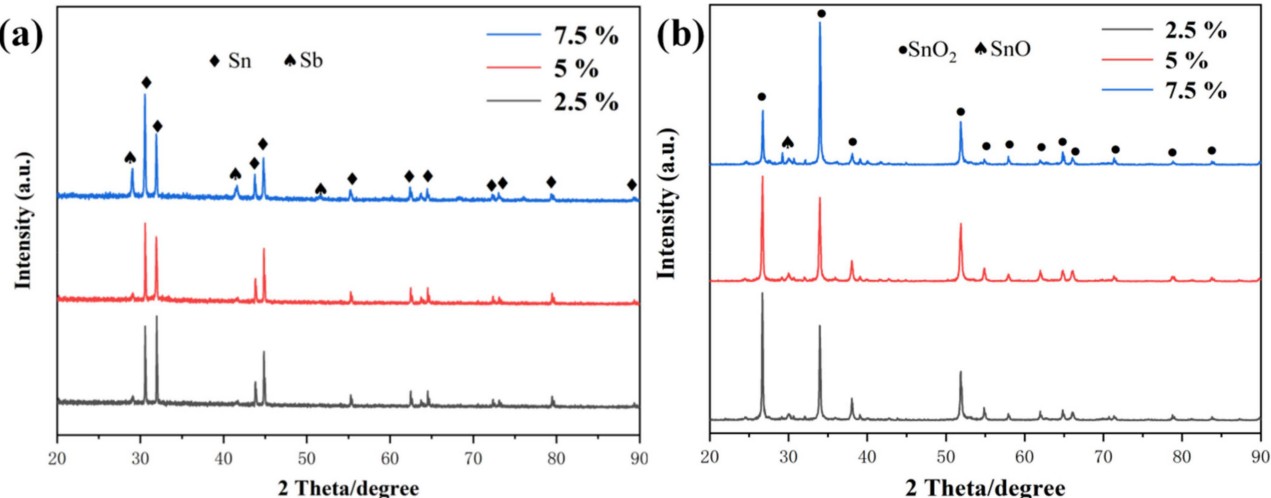

**Figure 9.** XRD diffractograms recorded of (**a**) Sn-Sb precoatings and (**b**) $SnO_2$-Sb coatings prepared at different Sb concentrations.

## 4. Conclusions

This study examined how the properties of $SnO_2$-Sb coatings are affected by different preparation parameters. Sn-Sb coatings were electrodeposited and then oxidized into $SnO_2$-Sb coatings through an annealing treatment. Increasing the current density altered the surface roughness of the prepared Sn-Sb precoatings, but the annealing treatment did not significantly alter the primary surface features of the $SnO_2$-Sb coatings. The suitable current density of 30 mA/cm$^2$ created a rough, active coating surface without a decrease in the coating thickness.

Our study's proper annealing temperature of 600 °C transformed Sn-Sb precoatings into $SnO_2$-Sb coatings and achieved an excellent coating quality. The change in Sb content affected the morphologic features of the prepared $SnO_2$-Sb coatings but did not vary the cassiterite $SnO_2$ phase in the mixed oxide coatings.

**Supplementary Materials:** The following supporting information can be downloaded at: https://www.mdpi.com/article/10.3390/coatings13050866/s1, Figure S1: The cross-sectional images taken on the samples made by different method: the plate sample was directly cut and vertically observed without polishing or grounding.

**Author Contributions:** Conceptualization, Z.H. and Y.W.; methodology, C.Y.; validation, Z.H.; formal analysis, Y.W.; investigation, C.Y.; resources, Z.H.; data curation, J.L.; writing—original draft preparation, Z.H.; writing—review and editing, C.Y., Z.M. and Y.W.; funding acquisition, Z.H. All authors have read and agreed to the published version of the manuscript.

**Funding:** This work was supported by the Natural Science Foundation of Jiangsu Province under Grant No. BK20201008.

**Institutional Review Board Statement:** Not applicable.

**Informed Consent Statement:** Not applicable.

**Data Availability Statement:** Not applicable.

**Conflicts of Interest:** The authors declare no conflict of interest.

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
