# Peer review of "Effects of Preparation Parameters on the Structural and Morphologic Properties of SnO2-Sb Coatings"

_coatings, doi:10.3390/coatings13050866_

Round 1

Reviewer 1 Report

The authors have submitted an article on Sn-Sb and SnO2-Sb coatings on titanium mesh. Altough the experimental procedure looks interesting, the paper lacks novelty as similar research was done previously and consists only of two analytical techniques: SEM coupled with EDS and XRD. Therefore, very basic assumptions can be derived from limited data. I do not recommend the proposed submission for Coatings journal. Authors should redesign their article and improve it before their next submission.

1. While the article emphasizes the influence of different synthesis parameters, no direct comparison of the results achieved in the manuscript to the existing work is made.

2. The SEM images of the cross-sections should be taken again because the ones that were given aren't good enough to draw the conclusions that were suggested.

3. XPS analysis could provide more insight into the formed Sn and Sb compounds.

4. What is the explanation of the phenomenon where the Sb under heat treatment at 600 or 700 °C for such a long time does not form oxides? This aspect could be discussed with the XPS results. 

5. It is difficult for the reader to distinguish the differences between pre-coatings and coatings, therefore, some additional information should be provided in the experimental section.

Author Response

Dear editor and reviewer:

Thank you for your careful review work on our submission. We have already revised and responded to all comments based on reviewers’ comments, and the primary changes are highlighted in the revised manuscript. The related amendments and responses to the reviewers’ comments are listed as below.

Response to reviewer #1:

  1. While the article emphasizes the influence of different synthesis parameters, no direct comparison of the results achieved in the manuscript to the existing work is made.

Response: Thanks for your comment. We have included several papers focusing on the influence of parameters on the SnO2-Sb coatings, in order to make comparison between our results and previous work.

  1. The SEM images of the cross-sections should be taken again because the ones that were given aren't good enough to draw the conclusions that were suggested.

Response: Thanks for your comment. We have repeated the SEM observation, but similar results were obtained. In our opinion, the cross-sectional images give essential information of coating thickness and uniformity, which matches the XRD and surface SEM observations. This provides strong proof to support our statement in the manuscript.

  1. XPS analysis could provide more insight into the formed Sn and Sb compounds.

Response: Thanks for your comment. The XPS analysis has been included in Fig. 4, which provides necessary information to study the surface state for the coatings. The oxide form of Sb and Sn are detected on the surface, which matches other characterization tests. Moreover, we found that the different current density causes slight change on the ratio of Sb3+/Sb5+ for the coating surface. This provides significant information to study the surface chemistry in electrocatalytic reactions.

  1. What is the explanation of the phenomenon where the Sb under heat treatment at 600 or 700 °C for such a long time does not form oxides? This aspect could be discussed with the XPS results. 

Response: Thanks for your comment. Actually, we did not find Sb after the treatment for 600 or 700 °C. Only a minor peak of Sb is found after the treatment of 500 °C as depicted in Fig. 5, which could result form the low heating temperature. During the XRD tests, the X ray may penetrate several micrometers depending on the nature of testing materials and testing method. Therefore, it is possible to detect a small amount of Sn existing in the coating, which probably comes from the inner part of the coating that haven’t been oxidized.

  1. It is difficult for the reader to distinguish the differences between pre-coatings and coatings; therefore, some additional information should be provided in the experimental section.

Response: Thanks for your comment. Sorry for our negligence. We have rewritten some sentences in the experimental section and make it clearer to tell the difference between precoating and coatings.

Reviewer 2 Report

The title should be changed to reflect how the research is perceived in the main text.

The abstract part should be a brief elaboration of the background, and then put forward your own methods for the study, and finally a brief summary of the experimental part with findings.

Author Response

Dear editor and reviewer:

Thank you for your careful review work on our submission. We have already revised and responded to all comments based on reviewers’ comments, and the primary changes are highlighted in the revised manuscript. The related amendments and responses to the reviewers’ comments are listed as below.

Response to reviewer #2:

The abstract part should be a brief elaboration of the background, and then put forward your own methods for the study, and finally a brief summary of the experimental part with findings.

Response: Thanks for your comment. We have re-organized the abstract part to be more coherent and readable, based on the reviewer’s comment.

Reviewer 3 Report

Review report (Coatings-2294793)

He et al. investigated the effect of three key parameters such as current density, annealing temperature, and elemental concentration. of structural and morphological properties of SnO2-Sb thin film. The effect was analyzed by SEM, EDX, and other characterization techniques. This manuscript is well-written and organized well. However, there are still some drawbacks that need to be improved. This paper is acceptable in the esteemed journal Coatings after a few comments and answers some of the important questions. Hence my recommendation is to publish after major revision. Here are the comments-

1.      The manuscript lacks the background of a solid literature survey, for example, reference no. 12-22 are directly related to SnO2-Sb thin film on the substrate but they are just cited, you are requested to discuss the results and findings from their study.

2.      Could you please more elaborate what is the impact of electrodeposition parameters on the coatings as you have said in lines 55-56? In-depth research is still very 55 much needed for the precise impact of these crucial electrodeposition parameters on SnO2- 56 Sb coatings [20-22].

3.      The effect of synthesis parameters on structural and physical properties is crucial to mention however the literature is so limited to a few parameters.

4.      The other key parameters such as microwave power, deposition time, pH, and deposition temperature could give a better understanding and support the contribution of this research hence, you are requested to discuss more results in a separate paragraph in the introduction section. A few examples of literature are suggested below-

https://doi.org/10.1007/s11082-020-02535-x

https://doi.org/10.1504/IJSURFSE.2018.090051

https://doi.org/10.1016/j.matlet.2015.04.074

https://doi.org/10.1016/j.matchemphys.2019.122277

5.       Please highlight the novelty of your work clearly in the last paragraph of the introduction section.

6.       In section 2.1, the grade and quality of a few chemicals and materials are given, please provide other chemical information as well.

7.       There is a significant decrease in Sb content after oxidization as can be seen in the results provided in EDX, what is the reason, it will be better if it could be explained in the current density section.

8.       Line 131-133, the authors claim that the XRD data is in agreement with the JCPDS card but no information is provided in the text or n the figure. Please provide a reference and JCPDS card no.

9.       No crystallite size is calculated from the XRD data. If calculated, then mention the value in the text and abstract as well.

10.   None of the IR and UV visible characterizations have been done.

11.   The effect of parameters on the morphology is the main concern of this article but the size and the particles or thickness of the film are not measured and mentioned in the articles. If calculated, then mention the value in the text and abstract as well.

12.   Is it possible to measure thin film thickness and provide information on key parameters' effect on thin film thickness?

13.   The effect of annealing temperature on the morphology of thin film is obvious but surprisingly not a single reference is cited. Below are some pieces of literature which are in agreement with the results. https://doi.org/10.1007/s10904-020-01646-y

https://doi.org/10.1016/j.matpr.2019.06.651

 10.1088/2053-1591/ab6122

14.    It is noted that figure 1 is just explained without mentioning fig.1a,1b, and so on. Similarly, 2a and 2b and so on. The same is the case in figure 3. Also, see there is sequence is wrong in figure 4, see line 157, it is supposed to be figure 4 a and d.

15.   The caption of figure 5 is confusing, there is no a, b, c,d,e, and f in figure 5.

16.   There is no d,e, and f in figure 6. Please clear up this confusion.

17.   There is no table in the paper and it is written as table 5 (line 183) and table 8 (line 228).

18.   Improve the abstract and conclusion after revising the manuscript based on the required suggestions.

19.   Please revise your manuscript thoroughly, spelling and grammar check is required.

Author Response

Response to reviewer #3:

  1. The manuscript lacks the background of a solid literature survey, for example, reference no. 12-22 are directly related to SnO2-Sb thin film on the substrate but they are just cited, you are requested to discuss the results and findings from their study.

Response: Thanks for your comment. The introduction section is revised, and more sentences are added to introduce previous reports on SbO2-Sb coatings further. The changed part has been highlighted in blue.

  1. Could you please more elaborate what is the impact of electrodeposition parameters on the coatings as you have said in lines 55-56? In-depth research is still very 55 much needed for the precise impact of these crucial electrodeposition parameters on SnO2- 56 Sb coatings [20-22].

Response: Thanks for your comment. The introduction section is revised based on the comments. We have further illustrated the reported effects of electrodeposition parameters on the prepared coatings to make the introduction section more informative.

  1. The effect of synthesis parameters on structural and physical properties is crucial to mention however the literature is so limited to a few parameters.

Response: Thanks for your comment. The introduction section is revised based on the comments. We have further illustrated the effects of other reported electrodeposition parameters on the prepared coatings to make the introduction section more informative.

  1. The other key parameters such as microwave power, deposition time, pH, and deposition temperature could give a better understanding and support the contribution of this research hence, you are requested to discuss more results in a separate paragraph in the introduction section. A few examples of literature are suggested below-

https://doi.org/10.1007/s11082-020-02535-x

https://doi.org/10.1504/IJSURFSE.2018.090051

https://doi.org/10.1016/j.matlet.2015.04.074

https://doi.org/10.1016/j.matchemphys.2019.122277

Response: Thanks for your comment. These reports provide insightful details for the possible readers. In the revised manuscript, we have properly cited them and make the manuscript more readable and coherent.

  1. Please highlight the novelty of your work clearly in the last paragraph of the introduction section.

Response: Thanks for your comment. We rewrite the last paragraph to tell the difference and novelty of our study.

  1. In section 2.1, the grade and quality of a few chemicals and materials are given, please provide other chemical information as well.

Response: Thanks for your comment. We have implemented the suggested changes.

  1. There is a significant decrease in Sb content after oxidization as can be seen in the results provided in EDX, what is the reason, it will be better if it could be explained in the current density section.

Response: Thanks for your comment. We have implemented the suggested changes.

  1. Line 131-133, the authors claim that the XRD data is in agreement with the JCPDS card but no information is provided in the text or n the figure. Please provide a reference and JCPDS card no.

Response: Thanks for your comment. We have added the standard JCPDS number in the manuscript when first introducing the diffractograms.

  1. No crystallite size is calculated from the XRD data. If calculated, then mention the value in the text and abstract as well.

Response: Thanks for your comment. We actually tried to calculate the grain size based on the XRD diffractograms. However, textured crystal plane are shown in the data and the diffractograms show different intense peak in the manuscript. It is not accurate to do the calculations. Still, we can roughly observe the variation trend for the grain size based on the XRD and SEM data.

  1. None of the IR and UV visible characterizations have been done.

Response: Thanks for your comment. We now include the surface XPS analysis for the prepared samples in Fig. 5, which provides more accurate information for the surface features. The related description can be found in the paragraphs just near Fig. 5.

  1. The effect of parameters on the morphology is the main concern of this article but the size and the particles or thickness of the film are not measured and mentioned in the articles. If calculated, then mention the value in the text and abstract as well.

Response: Thanks for your comment. We now add several sentences roughly estimate the coating thickness and particle size in the text and abstract.  

  1. Is it possible to measure thin film thickness and provide information on key parameters' effect on thin film thickness?

Response: Thanks for your comment. We now estimate the coating thickness and include the descriptions in the manuscript.

  1. The effect of annealing temperature on the morphology of thin film is obvious but surprisingly not a single reference is cited. Below are some pieces of literature which are in agreement with the results. https://doi.org/10.1007/s10904-020-01646-yhttps://doi.org/10.1016/j.matpr.2019.06.65110.1088/2053-1591/ab6122

Response: Thanks for your comment, and we feel sorry for our negligence. The paper is correctly cited in the manuscript to give more information for the readers.  

  1. It is noted that figure 1 is just explained without mentioning fig.1a,1b, and so on. Similarly, 2a and 2b and so on. The same is the case in figure 3. Also, see there is sequence is wrong in figure 4, see line 157, it is supposed to be figure 4 a and d.

Response: Thanks for your comment. We carefully checked manuscript, and add the necessary description as the reviewer suggested.

  1. The caption of figure 5 is confusing, there is no a, b, c,d,e, and f in figure 5.

Response: Thanks for your comment. We carefully checked manuscript, and add the necessary description as the reviewer suggested.

  1. There is no d,e, and f in figure 6. Please clear up this confusion.

Response: Thanks for your comment. We carefully checked manuscript, and add the necessary description as the reviewer suggested.

  1. There is no table in the paper and it is written as table 5 (line 183) and table 8 (line 228).

Response: Thanks for your comment and sorry for our negligence. It is actually a typo when writing the manuscript and we delete the mentioned typos.

  1. Improve the abstract and conclusion after revising the manuscript based on the required suggestions.

Response: Thanks for your comment. We rewrite the conclusion and abstract part to make it more coherent and informative.

  1. Please revise your manuscript thoroughly, spelling and grammar check is required.

Response: Thanks for your comment. We carefully revised the manuscript and have it checked by a native speaker. We now believe it meets the publishing standard as a academic report.

Round 2

Reviewer 3 Report

Authors have done all the enquiries, therefore the manuscript now can be accepted.

Author Response

Thanks for the reviwer's insightful comments that helps a lot in improving our manuscript.